# TINKERING WITH BLACK BOXES: COUNTERFACTUALS UNCOVER MODULARITY IN GENERATIVE MODELS

## ABSTRACT

Deep generative models such as Generative Adversarial Networks (GANs) and Variational Auto-Encoders (VAEs) are important tools to capture and investigate the properties of complex empirical data. However, the complexity of their inner elements makes their functionment challenging to assess and modify. In this respect, these architectures behave as black box models. In order to better understand the function of such networks, we analyze their modularity based on the counterfactual manipulation of their internal variables. Our experiments on the generation of human faces with VAEs and GANs support that modularity between activation maps distributed over channels of generator architectures is achieved to some degree, can be used to better understand how these systems operate and allow meaningful transformations of the generated images without further training.

## 1 INTRODUCTION

Deep generative models have proven powerful in learning to design realistic images in a variety of complex domains (handwritten digits, human faces, interior scenes). Complex neural architectures are now used to learn complex empirical data distributions by designing a non-linear function mapping a latent space to the space observations. In particular, two distinct approaches have recently emerged as state of the art: Generative Adversarial Networks (GANs) (Goodfellow et al., 2014), and Variational Autoencoders (VAEs) (Kingma and Welling, 2013; Rezende et al., 2014). Such architectures relate to a question that work in Neuroscience and Computer vision have long since tried to address: the relation between an observed scene and its high level internal representation. This has been framed using two objects: (1) the mapping of a 3D scene to its perceived (2D) image, called *forward optics*, (2) the converse mapping, called *inverse optics*, (see e.g. (Kawato et al., 1993)). Many computer vision algorithms have relied on *inverse graphics* approaches that model both forward and inverse optics simultaneously (Kulkarni et al., 2015). In recent years, emphasis has been put on producing compact descriptions of the scene in terms of high level features reflecting a *disentangled* latent representation that can be mapped back to the image. However, the fundamental asymmetry between the structure of original forward and inverse optics mappings has received less attention. A key difference is that forward optics can be concisely described with a restricted set of equations taking into account physical parameters of the scene, while inverse optics does not have an explicit form and relies heavily on prior assumptions to be solved numerically. The simplicity of the forward optics may allow an agent to efficiently manipulate and update internal representations, for instance to plan interactions with the outside world, following a *predictive coding principle* (Rao and Ballard, 1999). This supports that modularity of generative models should be assessed and enforced in order to understand and manipulate representations.

Achieving this aim for deep architectures is challenging, because they mostly behave as black boxes, making it difficult for users to interact with the generative process. Indeed, we can act on how the network is trained (e.g. the optimized objective), what it learns to generate, but not on *how* the learned generative process operates. For example, to use a face generator to create a face combining the eyes of one generated face with remaining features of another one may be achieved by either additional training or complex manipulation of the network's input or output. Directly influencing the generative process learned by the network on the other hand is made difficult due to the complexity of the function class entailed by the networks' non-linearities and high dimensional parameter space. To grasp the properties of such a system, a possible approach is to intervene on parts of the architecture that implements the generative function. Ideally, the effect of such interventions on the output would

be interpretable. This suggests we should uncover a modular structure in those architectures, such that each part of a network can be assigned a specific function.

In this paper, we propose that modularity can be quantified and exploited in a causal framework to infer whether modules within the architecture can be further disentangled. This hypothesis relies on the general principle of *Independence of Mechanisms* stating that the various mechanisms involved in generating the observed data can be modified individually without affecting each other (Peters et al., 2017). It has been recently demonstrated that this principle can be applied to generative models encountered in machine learning Besserve et al. (2018). One key aspect of causality frameworks is to allow evaluating with counterfactuals how the outcome of a observed system would have changed, provided some variables would have taken different values. We use such counterfactuals to assess the role of specific internal variables in the overall functioning of trained deep generative models and uncover the modular structure of these systems. We start by introducing this perspective formally with the notion of intrinsic disentanglement, and show that it extends the classical notion of disentangled representation investigated in the deep learning literature. Then, we introduce tools to analyze this disentanglement in existing systems. We show empirically how VAEs and GANs trained on a human face dataset express a form of modularity with intermediate activation maps responsible for encoding different parts of the generated images.

**Related work.** The issue of interpretability in convolutional neural networks has been the topic of intensive investigation. Most of that research however has focused on discriminative neural networks, not generative ones. In the discriminative case, efforts have been made to find optimal activation patterns for filters (Zeiler and Fergus (2014),Dosovitskiy and Brox (2016)), to find correlation between intermediate feature space and data features (Fong and Vedaldi (2017),Zhang et al. (2017b)) or to disentangle patterns detected by various filters to compute an explanatory graph Zhang et al. (2017a). Furthermore, explicitly enforcing modularity in networks has been tried recently with Capsule networks architectures (Sabour et al. (2017)), although Capsule network explicitly separate the architecture in different modules before training. A more detailed overview can found in review Zhang and Zhu (2018). It is important to emphasize discriminative and generative processes differ significantly, and working on generative processes allows to directly observe the effect of changes in intermediate representations on the generated picture rather than having to correlate it back input images. The recent InfoGAN network (Chen et al. (2016)) and other works (Mathieu et al. (2016); Kulkarni et al. (2015); Higgins et al. (2017)) in disentanglement of latent variables in generative models can be seen as what we define as extrinsic disentanglement. As such, we believe our intrinsic disentanglement perspective should be complementary with such approaches and are not in direct competition. Finally our approach relates to modularity and invariance principles formulated in the field of causality, in particular as formalized by Besserve et al. (2018).

## 2 MODULARITY AS INTRINSIC DISENTANGLEMENT

### 2.1 CAUSAL GENERATIVE MODELS

In our aim to isolate a modular functional structure in deep networks, we first introduce a general formalism to perform interventions inside a generative network. We will rely on the notion of causal generative models to represent any latent variable model used to fit observational data. Causality entails the idea that discovered relationships between variables have some degree of robustness to perturbations of the system under consideration and as a consequence allows predicting interventions and counterfactuals. Causal models can be described based on Structural Equations (SEs) of the form

$$Y \coloneqq f(X_1, X_2, \cdots, X_N, \epsilon),$$

expressing the assignment of a value to variable $Y$ based on the values of other variables $X_k$, with possibly additional exogenous effects accounted for through the random variable $\epsilon$. This expression thus stays valid if something selectively changes on the right hand side variables, and accounts for the robustness or invariance to interventions and counterfactuals expected from causal models as opposed to purely probabilistic ones (see for example Peters et al. (2017); Pearl (2009)). Such SEs can be combined to build a Structural Causal Model made of interdependent modules to represent a more complex system, for which dependencies between variables can be represented by a directed acyclic graph $\mathcal{G}$. Let us use such structural model to represent our generator:

**Definition 1** (Causal Generative Model (CGM)). *A causal generative model $M = \mathbb{G}(P_{\mathbf{Z}}, \mathbf{S}, G)$ consists of a distribution $P_{\mathbf{Z}}$ over $K$ latent variables $\mathbf{Z} = (Z_k)$ that accounts for exogenous effects,*

*a collection $S$ of structural equations assigning endogenous random variables $\mathbf{V} = (V_k)$ and an output $I$ based on values of their endogenous or latent parents $\boldsymbol{Pa}_k$ in the directed acyclic graph $G$. We assume $I$ has no latent parent, such that it is assigned by at least two deterministic mappings: one using latent variables and one using endogenous variables*

$$I = g_M(\mathbf{Z}) = \tilde{g}_M(\mathbf{V})\,.$$

*We assume $P_{\mathbf{Z}}$ is such that the $Z_k$'s take their respective values on intervals $\mathcal{Z}_k \subset \mathbb{R}$, on which they have a non-vanishing density with respect to the Lebesgue measure, and are jointly independent. We define the* image *of $M$, $\mathcal{I}_M = g_M(\prod_{k=1..K} \mathcal{Z}_k)$, i.e. the set of all possible objects generated by $M$.*

The graphical representation of a CGM is exemplified on Fig. 1a. This definition essentially aligns with the classical definition of a probabilistic Structural Causal Model (SCM) (see e.g. Pearl (2009); Peters et al. (2017)) and describes rigorously the functional structure of feed-forward generative models. Note that following Pearl (2009, chapter 7), we can exploit the fact that SCMs can been first defined as deterministic (when exogenous variables are fixed), from which we can derive the mappings $g_M$ and $\tilde{g}_M$. Simply combining this deterministic object with a probability distribution on exogenous variables leads to the classical probabilistic SCM definition. CGMs however have a few specificities with respect to classical SCMs. First, the latent variables correspond to the so-called *exogenous variables* in structural equations, and contrary to classical causal inference settings, these variables are presumably "observed" since a practitioner can simply access any variables pertaining to the generative model she uses. Second, the classically called "observed variables", corresponding to the nodes of the graphical model, consist of two subsets: the output $I$ (that we consider here as a single multidimensional variable) and the *endogenous variables* which are essentially internal variables corresponding to intermediate computations between the latent input and the output. We leave open the granularity of the partitioning of internal values of the network into endogenous variables: one variable $V_i$ may for example represent the scalar activation of one single neuron, or one (multivariate) channel (for example in the context of convolutional layers: a 2D activation map). Finally, the image set $\mathcal{I}_M$ of the model is of particular importance for applications of generative models, as it should approximate at best the support of the data distribution we want to model. For example, if we want to generate images of human faces, $\mathcal{I}_M$ certainly should not cover the space of all possible RGB images, but live on a complicated subset of it (possibly with a manifold structure). Correctly fitting the generator parameters such that $\mathcal{I}_M$ precisely matches the support of target distribution can be seen as the key objective of such models (see e.g. Sajjadi et al. (2018)).

One benefit of the causal framework is to be able to define interventions and counterfactuals. We will use the following definition.

**Definition 2** (Interventional CGM and unit level counterfactual). *Consider a CGM $M = \mathbb{G}(P_{\mathbf{Z}}, \boldsymbol{S}, G)$. For a value $\boldsymbol{v}_0$ of a subset of endogenous variables $\mathbf{V}_{|\mathcal{E}}$ indexed by $\mathcal{E} = \{e_1, .., e_n\}$, we define the* interventional CGM $M_{\boldsymbol{v}_0} = \mathbb{G}(P_{\mathbf{Z}}, \boldsymbol{S}^{\boldsymbol{v}_0}, G^{\boldsymbol{v}_0})$ *obtained by replacing the structural assignments for $\mathbf{V}_{|\mathcal{E}}$ by the constant assignments $\{V_{e_k} := v_k\}_{e_k \in \mathcal{E}}$. Then for a given value $\boldsymbol{z}$ of the latent variables called* unit, *the* unit-level counterfactual *output is the deterministic output of $M_{\boldsymbol{v}_0}$ for input latent variable $\mathbf{Z} = \boldsymbol{z}$*

$$I_{\boldsymbol{v}}^{\mathcal{E}}(\boldsymbol{z}) = g_{M_{\boldsymbol{v}_0}}(\boldsymbol{z}).$$

*We will call the $(\mathcal{E}, \boldsymbol{v}_0)$-counterfactual the mapping*

$$I_{\boldsymbol{v}_0}^{\mathcal{E}} : g_M(\boldsymbol{z}) \mapsto I_{\boldsymbol{v}_0}^{\mathcal{E}}(\boldsymbol{z})$$

*We say $I_{\boldsymbol{v}_0}^{\mathcal{E}}$ is* faithful *to $M$ if for all $\boldsymbol{z}$, $I_{\boldsymbol{v}}^{\mathcal{E}}(\boldsymbol{z})$ belongs to $\mathcal{I}_M$.*

This essentially corresponds to the definition of counterfactual that can be found in (Pearl, 2014), with notations adapted to the context of CGMs defined above. Note that we restrict for simplicity interventional CGMs to assigning constant values to endogenous variables, however interventions and their ensuing counterfactuals can be generalized to more complex assignments without major difficulty (see e.g. Peters et al. (2017)). This definition is also in line with the concept of potential outcome (see e.g. Imbens and Rubin (2015)). In addition, Pearl (2009, chapter 7)) provided a detailed discussion regarding this connection between SCM and potential outcome frameworks. Faithfulness is a property we introduce in the context of generative models to take into account that not all interventions on internal variables of a generative model will result in an output that likely belongs to the learned data distribution. For example, assigning a large value to a neuron may saturate the

non-linearity of many downstream neurons, resulting in an artifactual output. For such reason, we will repetitively need to restrict a function's output values to be in the same set as their inputs (i.e. their codomain is included in their domain), and we will call *endomorphism* such function.

## 2.2 TWO FORMS OF DISENTANGLEMENT

The notion of model modularity that we propose to isolate in networks can take multiple forms. We introduce here a formal definition of the above concepts of disentangled representation. In order to relate the definition to the concrete examples that will follow, we consider without loss of generality that the generated variable $I$ is meant to be an image.

**Definition 3** (Extrinsic disentanglement). *A CGM $M$ is extrinsically disentangled with respect to endomorphism $T : \mathcal{I}_M \to \mathcal{I}_M$ and subset of latent variables $\mathcal{L}$, if there exists an endomorphism $T'$ of the latent variables such that for any image generated by a realization $\mathbf{z} = \mathbf{Z}(\omega)$ of the latent variables ($I = g_M(\mathbf{z})$):*

$$T(I) = g_M(T'(\mathbf{z})), \tag{1}$$

*where $T'(\mathbf{z})$ only affects values of components of $\mathbf{z}$ in $\mathcal{L}$. We call sparsity of the disentanglement the minimum size $n$ of the subset $\mathcal{L}$. We say $T$ is $n$-disentangled in $M$.*

Extrinsic disentanglement can be seen as a form of intervention on the CGM as illustrated in Fig. 1b. In this figure, we apply a transformation that affects only $Z_1$ (we thus abusively write $T'(Z_1)$), thus modifying descendant nodes, leading to a modified output $I' = T(I)$, which is by construction 1-disentangled. We can easily see that this definition is compatible with the intuitive concept of disentangled representation as used for example by Kulkarni et al. (2015) in the context of inverse graphics, where $T$ would correspond to a change in e.g. illumination of the scene, while $T'$ would simply shift the values of the sparse set of latent variables controlling it. More generally, we can easily verify the any endomorphism in the latent variables will induce a disentangled transformation. The non-trivial challenge addressed in previous work on disentangled generative models is to have such transformation reflect interpretable changes in the content of generated objects while keeping disentanglement very sparse. Instead, we extend such analysis to the inner elements of the network.

**Definition 4** (Intrinsic disentanglement). *A CGM $M$ is intrinsically disentangled with respect to endomorphism $T : \mathcal{I}_M \to \mathcal{I}_M$ and subset of endogenous variables $\mathcal{E}$ if there exists an endomorphism $T'$ such that for any image generated by a realization $\mathbf{z} = \mathbf{Z}(\omega)$ of the latent variables ($I = g_M(\mathbf{z}) = \tilde{g}_M(\mathbf{v})$),*

$$T(I) = \tilde{g}_M(T'(\mathbf{v})) \tag{2}$$

*where $T'(\mathbf{v})$ only affects values of endogenous variables in $\mathcal{E}$.*

An illustration of this second notion of disentanglement is provided on Fig. 1c, where the split node indicates that the value of $V_3$ is computed as in the original CGM (Fig. 1a) before applying transformation $T'$ to the outcome. Intrinsic disentanglement directly relates to a causal interpretation of the generative model and its robustness to perturbation of its subsystems. To justify it, consider the case of Fig. 1d, where the GCM has an unaccounted latent variable $Z_3$. This may be due to the absence of significant variations of $Z_3$ in the training set, or simply bad performance of the estimation algorithm. If the remaining causal structure has been estimated in a satisfactory way, and the full structure is simple enough, a change in this missing variable can be ascribed to a change in only a small subset of the endogenous nodes. Then the transformation $T'$ from the definition can be seen as a proxy for the change in the structural equations induced by a change in $Z_3$. Broadly construed, appropriate transformations pairs $(T, T')$ emulate changes of unaccounted latent variables, allowing to check whether the fitted causal structure is likely to be robust to plausible changes in the dataset.

The endomorphism assumption for $T$ is again central for generative models, as not every $T'$ will lead to an output that remains in the CGM's image. Interestingly, the above defined faithful counterfactuals are relevant examples.

**Proposition 1.** *For a CGM $M$, if the $(\mathcal{E}, \boldsymbol{v}_0)$-counterfactual $I_{\boldsymbol{v}_0}^{\mathcal{E}}$ is faithful, then $M$ is intrinsically disentangled with respect to the transformation $I_{\boldsymbol{v}_0}^{\mathcal{E}}$ and subset $\mathcal{E}$.*

*Proof.* Since the intervention is faithful, $I_{\boldsymbol{v}_0}^{\mathcal{E}}(\boldsymbol{z})$ belongs to $\mathcal{I}_M$ for all $\boldsymbol{z}$, such that T is an endomorphism of $\mathcal{I}_M$. Then the obvious choice of $T'$ is the intervention that transforms $\boldsymbol{v}$ by assigning $\boldsymbol{v}_0$ to the subset of variables indexed by $\mathcal{E}$ and leaving the remaining endogenous variables unchanged. $\square$

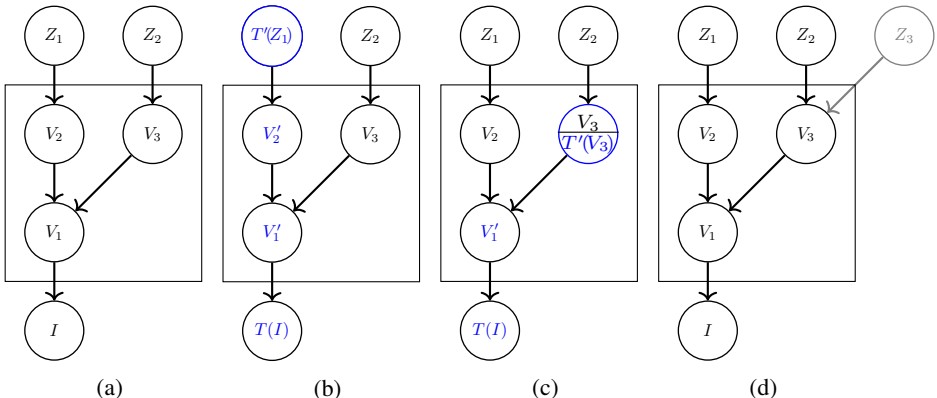

Figure 1: Graphical representation of CGMs. (a) Example CGM with 2 latent variables. (b) Illustration of extrinsic disentanglement with $\mathcal{L} = \{1\}$. (c) Illustration of intrinsic disentanglement with $\mathcal{E} = \{3\}$. (d) Illustration of unaccounted latent variable emulated by (c). Nodes modified by intervention on the graph are indicated in blue.

As it appears in the proof, Proposition 1 is a mere application of the previous definitions and aims at showing how the notions of disentanglement and interventions can articulated. The faithfulness assumption is not trivial to verify for non-trivial transformations in complex models, and can be seen as a way to define rigorously a form of modularity for the subset indexed by $\mathcal{E}$.

## 3 QUANTIFYING MODULARITY IN DEEP GENERATIVE MODELS

Quantifying modularity of a given GCM presents several challenges. State of the art deep generative networks are made of densely connected layers, such that modules cannot be identified easily beyond the trivial distinction between successive layers. In addition, analysis of statistical dependencies between successive nodes in the graph is not likely to help, as the entailed relations are purely deterministic. In addition, the notion of intrinsic disentanglement is specific to transformations $T$ and $T'$, and the relationship between these functions may be very complex. We propose a very general approach that avoids specifying such transformation pairs.

### 3.1 GENERATION OF HYBRID SAMPLES

Choosing the transformation to apply at a given module may be challenging without priors on the functional organization of the network. Indeed, values of activations in a given layer may be highly constrained by the precise arrangement of upstream synaptic weights and non-linearities. Therefore, as mentioned at the end of section 2.1, if we apply an arbitrary transformation to these activations without considering this issue, the internal variables may leave the domain they are implicitly embedded in as a result of the model's training, possibly resulting in the output of the generator leaving $\mathcal{I}_\mathcal{M}$.

To avoid a direct characterization of admissible values of internal variables, we rely on their variation across samples to intervene on the system. Indeed, if such values are generated from regular sampling of the latent variables, they will naturally belong to the right domain. We exploit this property to generate hybrid samples as follows. To simplify the presentation, we will take the example of a classic feed-forward multilayer neural network (with no shortcuts) and choose a collection of output values of a given layer $l$, corresponding to endogenous variables indexed by the subset $\mathcal{E}$. The hybridization procedure, illustrated in Fig. 2a, goes as follows. We take two independent samples of the latent variable $z_1$ and $z_2$, that will generate two *original* samples of the output $(g_M(z_1), g_M(z_2))$ (that we call *Original 1* and *Original 2*). We then record the tuple $v(z_2)$ gathering values of variables indexed by $\mathcal{E}$ when generating Original 2, and $\widetilde{v}(z_1)$ the tuple of values taken by all other endogenous variables at the output of the same layer $l$ when generating Original 1. If the choice of $\mathcal{E}$ identifies a modular structure, $\widetilde{v}(z_1)$ and $v(z_2)$ are assumed to encode different aspects of their corresponding generated images, such that one can generate a *hybrid sample* mixing these features by assigning all layer $l$ output values with the concatenated tuple $\widetilde{v}(z_1), v(z_2))$ and feeding it to the downstream part

of the generator network (i.e. the input of layer $l + 1$). The modularity assumption then justifies that these values still belongs to the range of typical layer $l$ activations for the unperturbed generator, and therefore that the hybrid output should still represent a reasonable sample of the learned distribution.

The above qualitative discussion can be made mathematically precise by using a counterfactual formulation wherein $\mathcal{E}$ is intervened on as in the context of Definition 2:

**Definition 5** (Counterfactual hybridization). *Given a CGM $M$, and two latent samples $z_1$ and $z_2$. Let $\mathcal{E}$ be a subset of endogenous variables and $v(z_2)$ the values assigned to these variables when the latent input is $z_2$. We define the $\mathcal{E}$-level hybridization of $z_1$ by $z_2$ as*

$$I^{\mathcal{E}}_{v(z_2)}(z_1).$$

Note this definition is more general than the above explanation as it allows for example interventions on a set of variables distributed on more than one layer.

## 3.2 COMPUTING INFLUENCE MAPS

The above counterfactual hybridization framework allows assessing how a given module (set of internal variables) affects the output of the generator. For this purpose we need to quantify the causal effect of counterfactuals. We assess such effect for a module indexed by $\mathcal{E}$ by repetitively generating pairs $(z_1, z_2)$ from the latent space, where both vectors are sampled i.i.d. independently of each other. We then generate and collect hybrid outputs following Definiton 5 for a batch of samples and use them to estimate an *influence map* as the mean absolute effect:

$$IM(\mathcal{E}) = \mathbb{E}_{z_2 \sim P(\mathbf{Z})} \left[ \mathbb{E}_{z_1 \sim P(\mathbf{Z})} \left[ \left| I^{\mathcal{E}}_{v(z_2)}(z_1) - I(z_1) \right| \right] \right] \tag{3}$$

where $I(z_1) = g_M(z_1)$ is the uninterveved output of the generator for latent input $z_1$. In equation 3, the difference inside the parenthesis can be interpreted as a *unit-level causal effect* in the potential outcome framework (Imbens and Rubin, 2015), and taking the expectation is analogous to computing the (population) *average treatment effect*. The main differences are: (1) that we take the entrywise absolute value of the unit-level causal effects, as their sign may not be consistent across units, (2) that we average the result over many "treatments" corresponding to different values of $z_2$.

While $IM$ has the same dimension as the output image, it is then averaged across color channels to get a single gray level pixel map. We also define a scalar quantity to quantify the magnitude of the causal effect, the *individual influence* of module $\mathcal{E}$, by averaging $IM$ across output pixels.

## 3.3 CLUSTERING GROUPS BASED ON INFLUENCE MAPS

A challenge with the above hybridization approach is to select the modules to intervene on, especially with networks containing a large amount of units or channels per layer. We propose a fine to coarse approach to extract such groups, that we will describe in the context of convolutional layers. First, we estimate *elementary influence maps* associated to each individual output channel $c$ of each convolutional layer of the network (i.e. we set $\mathcal{E} = \{c\}$ in equation (3)). Then influence maps are grouped by similarity to define modules at a coarser scale.

Representative influence maps for channels of convolution layers of a VAE trained on the CelebA face dataset (see result section) are shown on Fig. 2b and suggests channels are to some extent functionally segregated, with for example some influencing finer face feature (eyes, mouth,...) and other affecting the background of the image or the hair. This supports the idea that individual channels can be grouped into modules that are mostly dedicated to one particular aspect of the output image.

In order to achieve this grouping in an unsupervised way, we perform clustering of channels using their elementary influence maps as feature vector as follows. We first pre-process each influence map by: (1) performing a local averaging with a small rectangular sliding window to smooth the maps spatially, (2) thresholding the resulting maps at the 75% percentile of the distribution of values over the image to get a binary image. After flattening image dimensions, we get a (channel×pixels) matrix $S$ which is then fed to a non-negative matrix factorization algorithm with manually selected rank $K$, leading to the factorization $S = WH$. From the two resulting factor matrices, we get the cluster template patterns (the K rows of $H$ after reshaping to image dimensions), and the weights representing the contribution of each of these pattern to individual maps (encoded in $W$). Each influence map

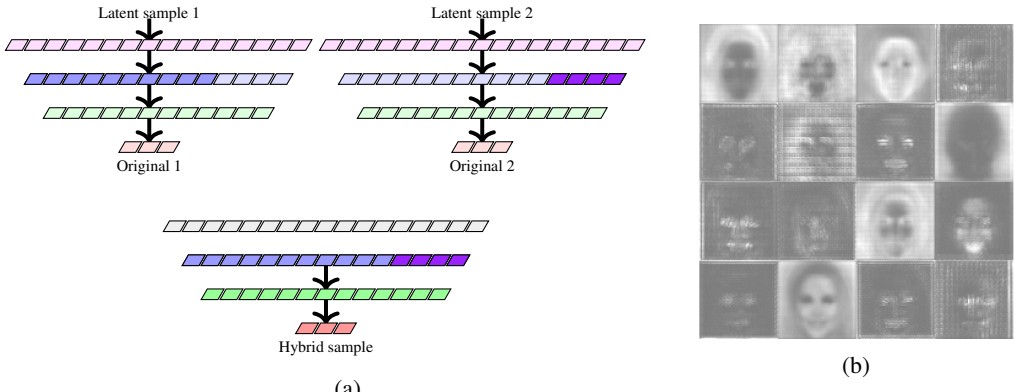

**Figure 2: Generation of influence maps.** (a) Principle of sample hybridization through counterfactuals. (b) Example of influence maps generated by a VAE on the CelebA dataset (lighter pixel indicate larger variance and thus stronger influence of the perturbations on that pixel).

is then ascribed a cluster based on which template pattern contributes to it with maximum weight. The choice of NMF is justify by its success in isolating meaningful parts of images in different components (Lee and Seung, 1999). However, we also compare our approach to the classical k-means clustering algorithm applied to the same preprocessed features.

## 4 EXPERIMENTS

We investigated our approach on real data in the form of the CelebFaces Attributes Dataset (CelebA)[1]. We used the official tensorlayer DCGAN implementation[2] and a plain $\beta$-VAE[3] (Higgins et al. (2017)). The general structure of the VAE is summarized in Fig. 4 and the DCGAN architecture is very similar. We separate the different layers in 4 levels indicated in Fig.4: coarse (closest to latent variables), intermediate, fine and image level (closest to the image). Complete architecture details are provided in the supplemental material. Unless otherwise stated, original samples generated by the VAEs result from the pass of a real image trough the encoder.

### 4.1 INFLUENCE MAP CLUSTERING AND HYBRIDIZATION IN VAES

We ran the full procedure described in previous section, comprised of influence maps calculation, clustering of channels into modules, and hybridization at the module level. Unless otherwise stated, hybridization procedures are performed by intervening at the output of the intermediate convolutional layer (indicated in Fig. 4). The results are summarized in Fig. 3. We observed empirically that setting the number of clusters to 3 leads consistently to highly interpretable cluster templates as illustrated in the figure, with one cluster associated to the background, one to the face and one to the hair. This observation was confirmed by running the following cluster stability analysis: we cut at random the influence maps in 3 subsets, and we use this partition to run the clustering twice on two thirds of the data, overlapping only on one third. The obtained clusters were then matched in order to maximize the label consistency (the proportion of influence maps assigned the same label by both clustering outcomes) on the overlapping subset, and this maximum consistency was used to assess robustness of the clustering across number of clusters. The consistency result are provided in Fig. 5 and show 3 clusters is a reasonable choice as consistency is large ($> 90\%$) and drops considerably for 4 clusters. Moreover, these result also show that the NMF-based clustering outperforms clustering with the more standard k-means algorithm. In addition, we also assessed the robustness of the clustering by looking at the cosine distance between the templates associated to matching clusters, averaged across clusters. The results, also provided in Fig. 5, are consistent with the above analysis with an average cosine similarity of .9 achieved with 3 clusters (maximum similarity is 1 for perfectly identical templates). Exemplary influence maps shown in Fig. 3 (center panel) reflect also our general observation: some

---

[1] http://mmlab.ie.cuhk.edu.hk/projects/CelebA.html

[2] https://github.com/tensorlayer/dcgan

[3] https://github.com/yzwxx/vae-celebA

maps may spread over image locations reflecting different clusters. However, being more selective by excluding maps that are not "pure" comes at the cost of reducing the influence of interventions on the resulting modules (result not shown).

Beyond choosing an optimal number of clusters, one can also assess how gathering coarser and coarser modules influences the magnitude of the causal effects of counterfactuals applied to them. To assess this, we computed the individual influence (average of influence maps across pixels) of modules associated to each clusters, when varying the number of clusters and hence the number of channels in each module. The results are shown on the right panel of Fig. 3, separating the analysis for the three last layers (layers 1, 2 and 3 corresponding respectively to the intermediate, fine and image level). We see that, as expected, the magnitude of the causal effect decreases with the number of clusters, because it increases with the number of elements per cluster, as illustrated by the linear regression fits shown on the bottom plot. In addition, magnitude of the causal effect is only weakly influenced by the choice of the layer that is intervened on. Overall, the results support the intuitive idea that the influence of a given module reflects the proportion of channels belonging to this module with respect to the total number of channels in the layer. As our layer size decreases exponentially from 64 to 32 from layer 1 to layer 2, this explains the difference in magnitude of individual influences at constant number of channels between these layers. However, we can also observe that the magnitude of causal effects is distributed heterogeneously across modules of the same size, and this heterogeneity is more striking in layer 3, leading to a poorer fit of the linear regression model. This suggests that causal influence is more irregularly distributed in layers closer to the output.

Interestingly, applying the hybridization procedure to the resulting 3 modules leads a replacement of the targeted features associated to the module we intervene on, as shown in Fig. 3 (right panel), while respecting the overall structure of the image (no discontinuity introduced). For example, on the middle row we see the facial feature of the *Original 2* samples are inserted in the *Original 1* image (show on the left), while preserving the hair. We found that the main observations in these hybridization experiments a rather consistent for reasonable choices of model parameters. In particular, the VAE model used in this experiment was making a trade-off between the sharpness of reconstructed images and the quality of images generated by sampling latent variables from the isotropic Gaussian prior. By decreasing the $\beta$ parameter, we can put more emphasis on the quality of reconstructed images. Performance of our procedure on such model ($\beta$ divided by 10) is shown in Fig. 6, where we can see better overall image quality, but a slightly more artificial hybridization with for example a slight overlay of the hair of both original images.

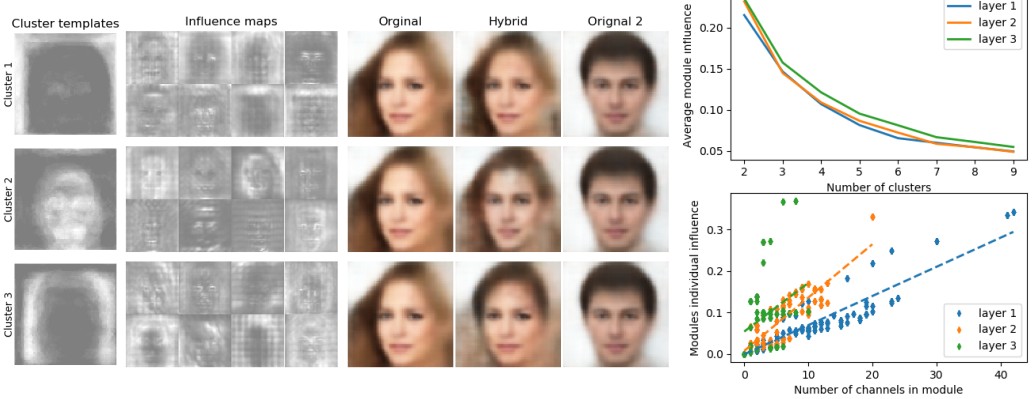

Figure 3: Left: Clustering of influence maps and generation of hybrid samples for a VAE trained on the CelebA dataset (see text).. Right: magnitude of causal effects. (Top: average influence of modules derived from clustering, as a function of the number of clusters. Bottom: individual influence of each modules, as a function of the number of channels they contain, dashed line indicate linear regression).

### 4.2 INFLUENCE MAP CLUSTERING AND HYBRIDIZATION IN GANS

We replicated the above approach for GANs on the CelebA dataset. The result shown in Fig. summarize the main differences. First, the use of three cluster seemed again optimal according to the stability of the obtained cluster templates. However, we observed that the eyes and mouth location were associated with the top of the head in one cluster, while the rest of the face and the sides of the image (including hair and background) respectively form the two remaining clusters. In this sense, the GAN clusters are less on par with high level concepts reflecting the causal structure of these images. However, such clustering still allows a good visual quality of hybrid samples.

Additional preliminary experiments were also conducted on the CIFAR10 dataset made up of 50000 pictures of animals and vehicles from 10 different categories (Fig. 8). Overall, the clustering procedure is more challenging to adjust, although several influence maps are clearly associated to objects in the foreground, and others to the background.

## 5 DISCUSSION

The purpose of this paper was to introduce a methodology to assess modularity in deep networks. Modularity may involve different aspects, and strongly depends on the nature of the modeled data. In this paper, we focused on features of the image that preferentially occur in specific parts of the generated images. This is a reasonable assumption for the CelebA dataset, given that the faces are spatially aligned. To some extent this is also true for the CIFAR10 dataset, where objects preferentially appear at the center of the image and the soil and sky in the background will be found at the bottom and top respectively. This approach may however have some limitations when looking at different datasets deprived from such spatial organization. In this case, capturing the structure of output variations induced by hybridization may require a more general approach. In principle, multidimensional technique such as Principal Component Analysis and non-linear generalizations may be able to characterize counterfactuals of each channels in order to further generate relevant modules following the steps described in the present work.

Another aspect that is left to further work is how to optimize modularity in deep generative networks. We believe that classical (extrinsic) disentanglement approaches will not help, as they focus on the input of the network without control on its internal structure. While current generative models seem to exhibit some amount of modularity, improving it may require specific learning objectives as well as an appropriate choice of architectures.

## 6 CONCLUSION

We propose approaching modularity in generative networks by intervening of their internal variables. To assess this notion of *intrinsic disentanglement*, we analyzed generative networks as Causal Generative Models and define procedure to characterize the role played by different groups of channels in these architectures. We found evidence for interpretable modules of internal variables in VAEs and GANs trained to generate images of human faces.

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

## APPENDIX

### NETWORK ARCHITECTURE

The general VAE architecture is presented in Fig. 4 for the particular case of the CelebA dataset. All other architectures used in our experiments follow the same general architecture with hyperparameters specified in Table 1

### NETWORK HYPERPARAMETERS

Default network hyperparameters are summarized in Table 1 (they apply unless otherwise stated in main text).

| Architecture | VAE CelebA | GAN CelebA | VAE cifar10 | GAN cifar10 |
|---|---|---|---|---|
| Nb. of deconv. layers/channels of generator | 4/(64,64,32,16,3) | 4/(128,64,32,16,3) | 3/(64,32,16,3) | 3/(64,32,16,3) |
| Size of activation maps of generator | (8,16,32,64) | (4,8,16,32) | (4,8,16) | (4,8,16) |
| Latent space | 128 | 150 | 128 | 150 |
| Optimization algorithm | Adam ($\beta = 0.5$) | Adam ($\beta = 0.5$) | Adam ($\beta = 0.5$) | Adam ($\beta = 0.5$) |
| Minimized objective | VAE loss (Gaussian posteriors) | GAN loss | VAE loss (Gaussian posteriors) | GAN loss |
| batch size | 64 | 64 | 64 | 64 |
| Beta parameter | 0.0005 | NA | 0.0005 | NA |

Table 1

### ONLINE SUPPLEMENTARY MATERIAL

Supplementary files are available at `https://www.dropbox.com/sh/4qnjictmh4a2soq/AAAa5brzPDlt69QOc9n2K4uOa?dl=0`

SUPPLEMENTAL FIGURES

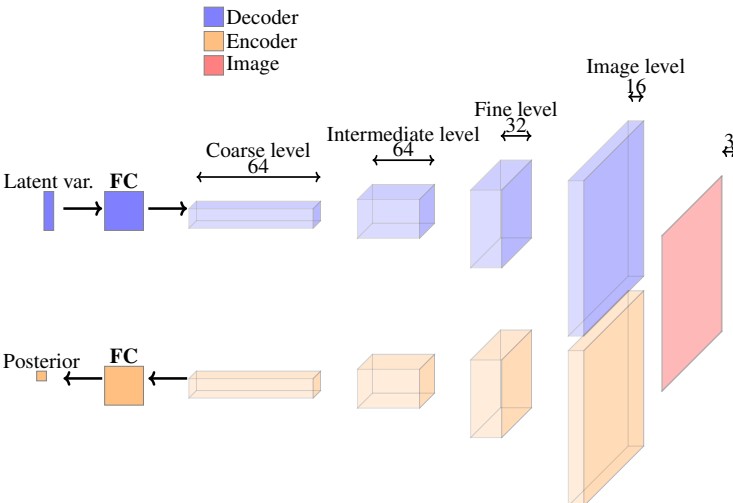

Figure 4: FC indicates a fully connected layer, $z$ is a 100-dimensional isotropic Gaussian vector, horizontal dimensions indicate the number of channels of each layer. The output image size is $64 \times 64$ (or $32 \times 32$ for cifar10) pixels and these dimensions drop by a factor 2 from layer to layer. (reproduced from (Radford et al., 2015).

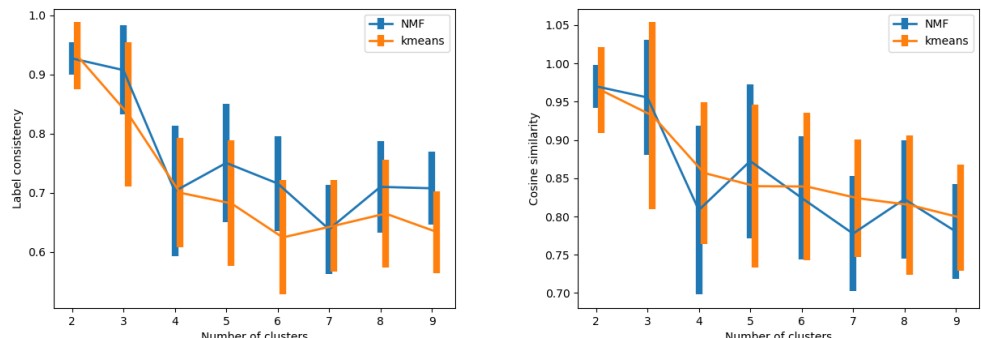

Figure 5: Label consistency (left) and cosine similarity (right) of the clustering of influence maps for the NMF and k-means algorithm. Errorbars indicate standard deviation across 20 repetitions.

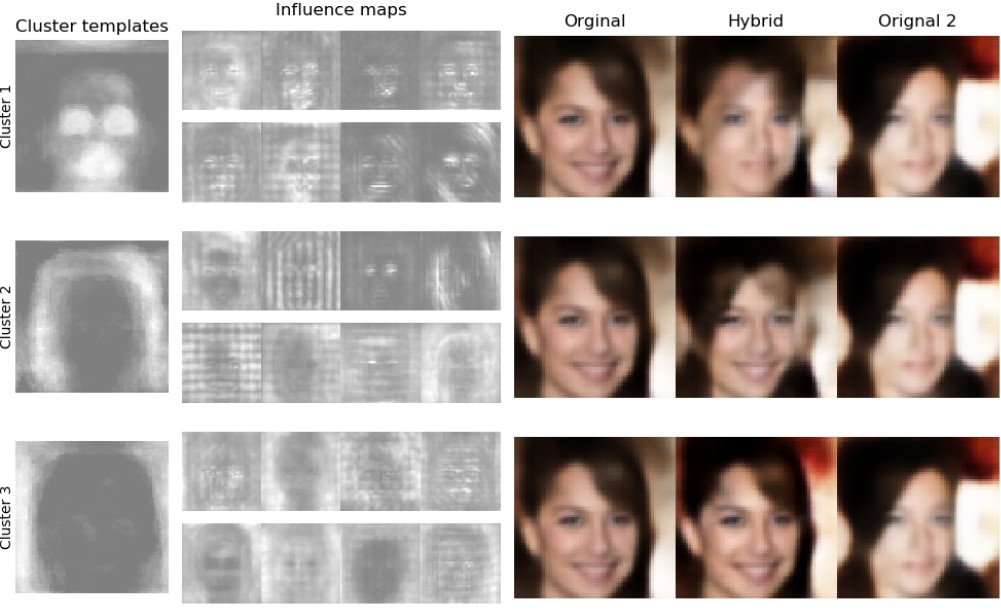

Figure 6: Clustering of influence maps and generation of hybrid samples for a VAE trained on the CelebA dataset (see text).

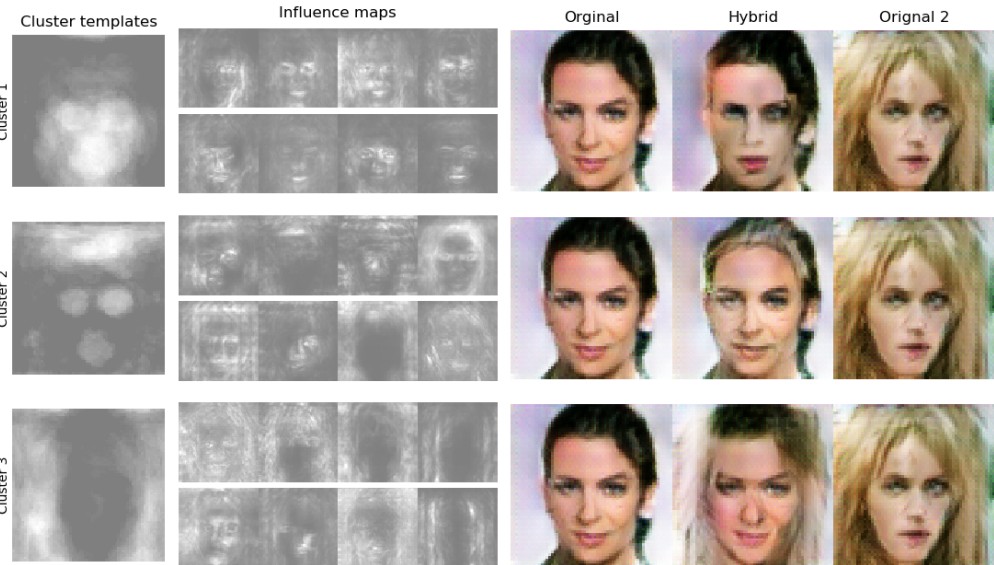

Figure 7: Clustering of influence maps and generation of hybrid samples for a GAN trained on the CelebA dataset (see text).

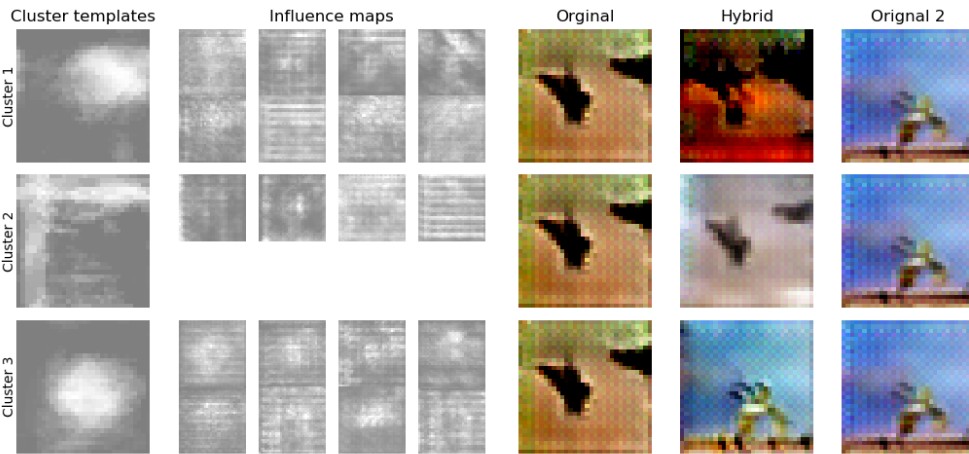

Figure 8: Clustering of influence maps and generation of hybrid samples for a GAN trained on the CIFAR10 dataset (see text).

