# OpenReview forum: "Tinkering with black boxes: counterfactuals uncover modularity in generative models"
_ICLR.cc/2019/Conference_

### Official Review · AnonReviewer3 · 2018-10-31
**The authors of this paper propose a method for assessing modularity in deep networks and more specifically on deep generative networks.**

**Rating:** 4
**Confidence:** 5

**Review:**

AFTER REBUTTAL:
I think that in its current version the paper is not yet ready for publication. Several issues have been raised by fellow reviewers as well. I think that they are not trivial and they regard key aspects like paper structure, quality of exposition and experimental analysis. I have detailed my initial opinion in response to the author request for more details.  I hope this will serve as useful  guidelines for improving the paper in the future.

------------
The method tackles the problem of interpretability that is a very important issue for usually black-box deep networks. Unfortunately it is not very clear how is the achieved. I have read several times the part explaining the influence maps and the clustering based on them and it still doesn't make a lot of sense to me. I think that part has to be better justified and exposed. Moreover, results do not support the claim which makes me doubt even more about how effective the method proposed actually is. In conclusion, I think that better exposition and more solid experimental analysis is needed.

Also please check some writing problems:

> Introduction:
"to acquire a generative function mapping a latent space (such as Rn)" >  difficult to read, rephrase.
"making it difficult to add human input" > confusing. What do you mean by human input? I assume you refer to having control to make decisions about design.

> Section 3.1
"the internal variable may leave the manifold it is implicitly embedded in as a result of the model’s training" : not clear, rephrase.

---

> ### Author Response · Authors · 2018-11-27
> **Clarifications about our approach**
>
> Dear Reviewer 3,
> We have rephrased the unclear sentences you pointed out, many thanks. Regarding the lack of clarity of the approach, we have considerably improved the explanation and rigorous formulation of our analysis in the revision. In particular, Definition 2 and 5 as well as equation (3) now describe in detail what is done. We would like to point out that what we are doing considerably differs from classical interpretability approaches, as it relies on changing variables inside the computational graph and assess how these changes modify the output of the generator. We claim that different meaningful aspects of the generated image can be intervened on independently in such a way, and our result on the CelebA support our claim surprisingly well. In particular, it was striking for us to see that intervening on different parts of the output image by acting the first convolutional layers (further from the image) was possible.

---

### Official Review · AnonReviewer1 · 2018-10-31
**Interesting concept but not ready for publication.**

**Rating:** 4
**Confidence:** 4

**Review:**

This paper proposes to examine generative adversarial networks by using counterfactual reasoning. The authors propose to examine modularity through the lens of interventions on the generative networks. After observing that the nodes within the generative network obey a deterministic relationship, they propose a proxy for intervention which takes samples and creates “hybrid” samples by replacing the activation output of one sample with the others. Given the vast number of nodes that exist within a generative network, the authors propose a heuristic for choosing the nodes to perturb.

I found the underlying premise of this paper to be very strong (identifying modularity in generative networks), however I think there is a substantial amount of work that should go into this paper before acceptance. While the authors begin by working within the framework of causal reasoning there is no mention of what the effect is that they are seeking to measure, i.e. what is the causal estimand here? The influence maps provide an intuitive answer to this, but not one that defines a clear estimand. I would like to see additional evaluation. The evidence provided largely leaves the reader to interpret results subjectively, rather than providing clear evidence. I was also uncomfortable with the selection on hyperparameters (3 clusters). It would be very nice to either have a selection criterion or show the sensitivity of the proposed methodology to other choices.

Overall, I think this is an interesting idea in a very important area, but one that is not quite ready for publication.

Some editorial comments:

The layout of this paper is slightly strange. After the introduction, the authors introduce the notion of disentanglement and lead with an example from optics. This motivation should either be moved to the introduction or removed. After the definitions the authors jump into a related work section that feels slightly disjointed from the previous section.

I found definition 1 to be abstruse. In addition there are a couple of typos that should be addressed (“consists in a distribution” → “consists of a distribution”). It is non-standard to lead with the latent variables. I think it makes for a much easier narrative to describe the observed variables and structure first, before carrying on to the latent variables. Additionally, I believe you are stating an observation made by Pearl (2001) that after observing the noise variable, relationships become deterministic. This is slightly non-obvious from the wording used (and is also missing the proper reference).

Parens are missing from the following citation:
“generative models encountered in machine learning Besserve et al. (2018).”  → “generative models encountered in machine learning (Besserve et al., 2018).”

---

> ### Author Response · Authors · 2018-11-27
> **Concerns addressed**
>
> Dear Reviewer 1, thanks to your feedback, we improve the organization and clarity of the paper, moreover we added more quantitative analysis to the results. We provide below answers to your main concerns.
> 1.	What is the causal estimand?
> Very good point, we clarified this in the revision by first defining unit-level counterfactuals in section 2 (Definition 2) and then introducing the hybridization operation as counterfactuals in section 3.1. Finally, the causal estimand (at the population level) is written in equation 1 and corresponds to the average absolute value of the unit level causal effect in the potential outcome framework.
> 2.	Justification for the number of clusters
>  Assessing the optimal number of clusters is a notoriously difficult and still debated problem in unsupervised learning. We addressed it by quantifying the consistency of the labelling provided by the clustering algorithm, when it is trained on different but overlapping datasets. One benefit of such approach is that this analysis can be applied to any clustering approach, which allowed us to compare the performance of the classical k-means algorithm with respect to our NMF based approach. This is described in the revised section 4.1, and the results depicted on Fig. 5 (in the appendix) suggest using NMF with 3 clusters is a reasonable choice as the consistency drops strongly for 4 clusters.
> 3.	Subjective interpretation of the results
> Assessing objectively the performance of generative models is also still largely debated in the field. We have however performed quantitative analysis in this revision by investigating in Fig. 3 the magnitude of the causal effect as a function of the size of the modules that we create with clustering and intervene on. The results are interesting as they exhibit to some extent a linear dependency between the causal effect and the size of the cluster, that tends to become more complex for layers closer to the image.
> 4.	Strange layout
> We reduced and moved the optics and related work sections to introduction.
> 5.	Abstruse Definition 1
> We rewrote the Definition 1 and provided more extensive explanations below, however we could not see an easy way to lead with observed variables, as in our analysis (including new definition 2 and 5 and proposition 1), the mapping from latent space to observations is central. As we added in the comments below Definition 1, the present context is quite different from classical causality settings as we are given the whole generator architecture, so every variables, included latent ones can be observed by the user. In that context, emphasizing the deterministic mapping from the latent to the output variables seemed more natural to us.
> 6.	Deterministic mapping in structural equations
> It is correct, the deterministic mapping plays a key role for our counterfactual analysis, and follows from the very definition of structural equation models. We clarified this after the Definition 1 and added reference to chapter 7 of Pearl, 2009, where structural equation are first introduced in a deterministic setting.

---

### Official Review · AnonReviewer2 · 2018-11-05
**causality based investigation of the modular structure of the deep generative model**

**Rating:** 6
**Confidence:** 3

**Review:**

The work provides a way to investigate the modular structure of the deep generative model. The key concept is the “distribute over channels of generator architectures”.
strong points:
1) using the causality to investigate the modular of the deep generative model.
2) the key concept is interesting and straightforward.
3) the observations in the experiments are interesting.

But I have the following concerns,
1) the concept of counterfactual is consistent with that in the causality context?
2) more details of the causal model of the deep learning are needed,
3) more details of section 3.1 and 3.2 are needed, especially why these processes are proper interventions?

---

> ### Author Response · Authors · 2018-11-27
> **Concerns addressed**
>
> Dear Reviewer 2, thanks to your comments we have made our counterfactual framework more precise. Here are concise replies to your concerns.
> 1)	Yes the concept is consistent with counterfactual as defined by Pearl and with the potential outcome framework of Rubin. We added Definition 2 and 5 in order to precisely define counterfactuals and hybridization as a special case. Essentially, we based our framework on unit level counterfactuals (Pearl, 2014) consisting in: a) assigning the distribution of latent variables to a deterministic value obtained on a single sample of the latent variables (called unit in this framework). 2) intervening on one or several variables of the causal model. In the case of hybridization, the intervention consists in assigning to the output values of a subsets of channels in a layer to the value they take for another sample of the latent variables.
> 2)	We updated Definition 1 and provided more explanations below. We now draw and explicit connection between disentanglement and interventions in the context of causal models with the newly introduced Proposition 1.
> 3)	We described now rigorously hybridization as an intervention in section 3.1, for which we also improved and simplified the explanation. We also improved section 3.2 by giving a mathematical definition of influence maps (equation 3), and connected it to causal effects computed in the potential outcome framework.

---

### Meta-Review · Area_Chair1 · 2018-12-09
**Interesting ideas but unclear presentation**

**Confidence:** 4
**Recommendation:** Reject

**Metareview:**

This paper explores an interpretation of generative models in terms of interventions on their latent variables.  The overall set of ideas seems novel and potentially useful, but the presentation is unclear, the goal of the method seems poorly defined, and the qualitative results (including the videos) are unconvincing.

I recommend you put work into factoring the ideas in this paper into smaller ones.  For instance, definition 1 is a mess.  I would also recommend the use of algorithm boxes.